# Reversible Thermo-Optical Response Nanocomposites Based on RAFT Symmetric Triblock Copolymers (ABA) of Acrylamide and *N*-Isopropylacrylamide and Gold Nanoparticles

**DOI:** 10.3390/polym15081963

**Published:** 2023-04-21

**Authors:** Nery M. Aguilar, Jose Manuel Perez-Aguilar, Valeria J. González-Coronel, Hugo Martínez-Gutiérrez, Teresa Zayas Pérez, Enrique González-Vergara, Brenda L. Sanchez-Gaytan, Guillermo Soriano-Moro

**Affiliations:** 1Chemistry Center, Science Institute, Meritorious Autonomous University of Puebla (BUAP), University City, Puebla 72570, Mexico; 2School of Chemical Sciences, Meritorious Autonomous University of Puebla (BUAP), University City, Puebla 72570, Mexico; 3School of Chemical Engineering, Meritorious Autonomous University of Puebla (BUAP), University City, Puebla 72570, Mexico; 4National Polytechnic Institute (IPN), Center for Nanosciences and Micro and Nanotechnologies, Luis Enrique Erro, Mexico City 07738, Mexico

**Keywords:** poly(*N*-isopropylacrylamide), polyacrylamide, gold nanoparticles, RAFT polymerization, triblock copolymers

## Abstract

The development of composite materials with thermo-optical properties based on smart polymeric systems and nanostructures have been extensively studied. Due to the fact of its ability to self-assemble into a structure that generates a significant change in the refractive index, one of most attractive thermo-responsive polymers is poly(*N*-isopropylacrylamide) (PNIPAM), as well as its derivatives such as multiblock copolymers. In this work, symmetric triblock copolymers of polyacrylamide (PAM) and PNIPAM (PAM_x_-*b*-PNIPAM_y_-*b*-PAM_x_) with different block lengths were prepared by reversible addition−fragmentation chain-transfer polymerization (RAFT). The ABA sequence of these triblock copolymers was obtained in only two steps using a symmetrical trithiocarbonate as a transfer agent. The copolymers were combined with gold nanoparticles (AuNPs) to prepare nanocomposite materials with tunable optical properties. The results show that copolymers behave differently in solution due to the fact of variations in their composition. Therefore, they have a different impact on the nanoparticle formation process. Likewise, as expected, an increase in the length of the PNIPAM block promotes a better thermo-optical response.

## 1. Introduction

Smart polymers are one of the most interesting materials due to the fact of their ability to modify some of their physical or chemical properties when they are subjected to certain external stimuli, such as mechanical forces and variations in pH or temperature [1,2,3,4,5,6]. Particularly, polymers whose hydrophilic–hydrophobic behavior is susceptible to temperature variations (i.e., thermo-responsive polymers) are rapidly increasing in their applicability for the development of novel materials. Their particular response, attributed to conformational changes that are energetically favored according to the hydrophobic effect and the Gibbs equation [7,8,9], makes them versatile materials, especially for biomedical and temperature monitoring applications [10,11,12,13]. Currently, Poly(*N*-isopropylacrylamide) (PNIPAM) and its copolymers are one of the most studied thermo-responsive polymers for this purpose [14,15,16,17].

PNIPAM displays a temperature-dependent water solubility. At room temperature, PNIPAM’s chemical structure favors hydrogen bonds between water and the polymer, making the polymer highly soluble in aqueous solution. However, as the temperature increases and passes a specific point, known as the lower critical solution temperature (LCST), hydrogen bonds break and the structure self-assembles into an insoluble globular conformation, producing a reversible change in the refractive index [18]. Similarly, diblock copolymers of PNIPAM can display a temperature-dependent amphiphilic behavior and self-assemble at a temperature above the LCST. Thus, these copolymers can adopt interesting morphologies in solution, such as vesicles, micelles, and even worms, depending on the chemical structure and length of each polymer block [19,20,21,22]. 

Multiblock PNIPAM copolymers, especially triblock copolymers, have even more complex behavior in solution, since the block order is an important factor in the self-assembly process [23]. It has been shown that the sequence of blocks, (e.g., ABA, BAB, and ABC) impacts on the size–morphology of the polymer array in solution [24,25] and can even favor the formation of a physically crosslinked network at a temperature above the LCST [26,27]. This versatility has boosted the ability of multiblock copolymers to be combined with other materials, particularly plasmonic nanoparticles, to design novel composites that can be used to design biosensors and drug delivery systems, among others [28,29,30]. Prompted by this, triblock copolymers of polyacrylamide-*b*-poly(N-isopropylacrylamide)-*b*-polyacrylamide (PAM_x_-*b*-PNIPAM_y_-*b*-PAM_x_), with different block lengths, were prepared in a two-step reversible addition−fragmentation chain-transfer polymerization process (RAFT). The use of a symmetrical trithiocarbonate compound as a transfer agent allowed to obtain a controlled ABA block order [31]; to date, there are few reports that have described the synthesis of PAM-*b*-PNIPAM copolymers by RAFT polymerization. The size of the blocks was varied by modifying the feeding of the PAM block, which acted as a macro-RAFT agent, and the NIPAM monomer. These copolymers were used as templates for the formation of gold nanoparticles, by a modified Turkevich reaction, to prepare nanocomposites with a reversible thermo-responsive behavior. Gold nanoparticles (AuNPs) are important nanostructures within the materials field because of their multiple properties, including the ability to couple their electron density with electromagnetic radiation [32,33]. This phenomenon causes the nanostructures to display different colors that strongly depend on their size, morphology, and, especially, the chemical environment that surrounds them. Therefore, the conformational variation of the PNIPAM block, at a certain temperature, impacts on the nanoparticles’ chemical milieu, generating a significant color change. The PAM blocks, on the other hand, allow AuNPs to strongly attach to the polymer due to the excellent affinity between these nanostructures and the AM primary amide [34]. 

## 2. Materials and Methods

### 2.1. Materials

Acrylamide monomer (AM; ≥99%), *N*-isopropylacrylamide (NIPAM; ≥97%), 4,4′-Azobis(4-cyanovaleric acid) (ACVA; ≥98%), 2,2′-Azobis(2-methylpropionitrile) (AIBN; ≥99%), gold(III) chloride solution (HAuCl_4_; ≥99.9%), and sodium citrate (≥99%) were purchased from Sigma-Aldrich, Toluca, Mexico. The RAFT agent 2,2′-(thiocarbonylbis(sulfanediyl))bis(2-methylpropanoic acid) was synthesized [35]. Soy lecithin and toluene (99.5%) were from Química Mercurio, Puebla, Mexico. All materials were used without further purification. 

### 2.2. Polyacrylamide Block (PAM) Synthesis

The polyacrylamide block was synthesized using the inverse emulsion polymerization method. Briefly, the continuous phase was prepared by dissolving soy lecithin (120 mg) and the previously synthesized trithiocarbonate compound (264 mg) in toluene (15 mL). The dispersed phase was prepared separately by dissolving AM (10 g) in water (13 mL) and then mixed with the continuous phase. After the homogenization of both phases under continuous stirring, the system was purged with a nitrogen flow, sealed, and heated. When the temperature reached 80 °C, ACVA (20 mg dissolved in 2 mL of toluene) was added to the system without disturbing the inert atmosphere. The reaction was carried out for 5 h. Finally, the PAM block was purified by precipitating it in methanol and then dried at 60 °C.

### 2.3. Triblock Copolymers’ Synthesis (PAM-b-PNIPAM-b-PAM)

The PAM block was used as a macro-RAFT agent to incorporate the PNIPAM block into the structure by solution polymerization. In this process, different proportions of the PAM block and NIPAM monomers were mixed in acetone (28 mL). The feeding percentages (percentages by weight) tested were 80–20, 50–50, and 20–80 of PAM-NIPAM (Table 1), respectively. Once both the PAM block and NIPAM monomer were added, the system was purged with a continuous nitrogen flow. Subsequently, it was heated, and when the temperature was kept constant at 85 °C, AIBN (16 mg dissolved in 2 mL of acetone) was added without disturbing the inert atmosphere. The reaction lasted 5 h. The triblock copolymers obtained were washed with chloroform and dried at 60 °C. 

### 2.4. AuNPs Nanocomposites Synthesis (PAM-b-PNIPAM-b-PAM-Au)

The nanocomposites were prepared by the Turkevich method slightly modified. In a typical Turkevich synthesis, the Au^3+^ ions, obtained from the dissociation of a gold salt (HAuCl_4_), are reduced to Au^0^ and stabilized by the addition of reducing agents such as sodium citrate [36]. In this work, a triblock copolymer was added to the reaction medium after incorporating the reducing agent (sodium citrate) following the synthesis procedure described below. Briefly, an aqueous solution of HAuCl_4_ (7 mL, 0.5 mM) was brought to a boil. Then, a solution of sodium citrate (18 mg dissolved in 1 mL) was added. After a minute and a half, the solution turned colorful, indicating the AuNPs’ formation. At this point, the copolymer selected in solution (0.5 g in 6 mL acetone) was added dropwise. After this addition, the reaction was stopped and cooled to room temperature. The same procedure was carried out with each of the previously synthesized block copolymers.

## 3. Characterization

Proton nuclear magnetic resonance (^1^H NMR) was performed using a Bruker Avance III 500 MHz NMR. The measurement was carried out in solution using 15 mg of each triblock copolymer dissolved in deuterated water (D_2_O).

The nanoparticle size was monitored by dynamic light scattering (DLS) using a ZEN3690 zetasizer, NanoZS90. The analyses were performed taking an aliquot of each nanocomposite solution.

The UV–Vis spectroscopy measurements were carried out on a Varian Cary 50 spectrophotometer (Agilent Technologies, Santa Clara, CA, USA) with a xenon lamp. The measurements were conducted with nanocomposite aqueous solutions using a quartz cuvette with a 1 cm path length.

The determination of the viscous molecular weight (M*_v_*) of the PAM block was carried out using an Ubbelohde viscometer and a water bath at a constant temperature of 24.7 °C. One hundred milligrams of the sample were used and diluted in 12.5 mL. The intrinsic viscosity was obtained by the Solomon–Ciuta equation. The Mark–Houwink parameters used were k (×10^3^) (mg/mL) = 68 and a = 0.66, as reported in the literature, considering a molecular weight range between 1 and 20 (×10^4^), since RAFT polymerizations frequently produce polymers with low molecular weights [37].

Scanning transmission electron microscopy (STEM) micrographs were acquired with 30 kV using a JSM-7800F scanning electron microscope. A drop of the nanocomposite solution to be analyzed was deposited on a copper grid and dried at 36 °C to “freeze” the self-assembly arrangements of the triblock copolymers at this temperature. 

The turbidimetry measurements were performed using a Hach^®^ 2100AN Laboratory Turbidity Meter (Loveland, CO, USA), which was EPA compliant. The analyses were conducted considering all of the solutions resulting from the nanocomposites. To observe the changes in the refractive index of the nanocomposites with respect to the LCST, the solutions were measured at room temperature and after heating with stirring at 36 °C.

## 4. Results and Discussion

As described above, these triblock copolymers were prepared by a two-step RAFT polymerization process, as shown in Figure 1. For the first part (i.e., the formation of the first block), the inverse emulsion polymerization method was used [34]. This technique takes place in a heterogeneous milieu constituted by two immiscible phases, namely, an organic solvent that constitutes the continuous phase and a < n aqueous dispersed phase. The continuous phase contains the initiator, as well as the transfer agent, while the aqueous phase dissolves the monomer. During synthesis, the interaction between both phases was facilitated by a surfactant, creating micelles with the dispersed phase in the interior. Thus, throughout the reaction, all different stages of this polymerization process, including chain growth, occurred within those micelles. The ABA block sequence was achieved using a symmetrical trithiocarbonate compound such as a RAFT agent during the PAM block synthesis [31]. The symmetry of this agent suggests that trithiocarbonate substituents, represented in pink in Figure 1, provide two good homolytic leaving groups [38]. Thus, the fragmentation of these groups promoted the simultaneous PAM block growth on both sides of the trithiocarbonate, placing this functional group in the middle of the chain [39], providing the resulting species with the capacity to generate a highly symmetrical ABA-type block copolymer, as shown in Figure 1a. The RAFT agent was characterized by proton and carbon nuclear magnetic resonance (^1^H and ^13^C NMR), and the PAM block was characterized by ^1^H NMR, as seen in the Appendix A. The incorporation of the transfer agent into the PAM block was verified by UV-Vis, as shown in Appendix A. Since RAFT agents present a characteristic yellow coloration due to the presence of sulfur atoms from the trithiocarbonate, the resulting polymer is colored, indicative of the incorporation of the transfer agent and its presence throughout the reaction [40].

The chemical functionality of the PAM block allows this structure to act as a macro-RAFT agent to grow the PNIPAM block with high symmetry. Therefore, the PAM block fragments in the same way as the RAFT agent, so the NIPAM monomer adds to the side segments of the trithiocarbonate group, as shown in Figure 1b. Moreover, the triblock copolymers still maintain the yellow coloration due to the presence of the transfer agent, i.e., PAM block.

The structural characterization of these triblock copolymers was carried out by FTIR spectroscopy and proton nuclear magnetic resonance (^1^H NMR). According to FT-IR, the three copolymers displayed the characteristic bands of both the PAM and PNIPAM blocks, as Figure 2 shows. However, the main difference between the spectra was the band intensity of each block due to the fact that each copolymer was composed of different proportions. The polyacrylamide blocks exhibited the N–H_2_ vibrations of the primary amide at 3400 cm^−1^ and 3200 cm^−1^. The band of the CH_2_ backbone was at 2980 cm^−1^. In addition, the C=O stretching vibration bands of the amide group were at 1645 cm^−1^ and 1603 cm^−1^. The block of PNIPAM, on the other hand, presented the N-H vibration of the secondary amide at 3294 cm^−1^. The vibrations of the isopropyl group were located at 2972 and 2926 cm^−1^. In addition, the bands of the C=O stretching and N bond to the isopropyl group of the secondary amide were at 1641 cm^−1^ and 1536 cm^−1^.

The representative chemical shift signals of both blocks, PAM and PNIPAM, are shown by the uppercase and lowercase letters, as seen in Figure 3. The ^1^H NMR spectra show that the intensity of the signals were proportional to the PAM block and NIPAM monomer feed. This confirms that the triblock copolymers had a different chemical composition due to the variation of the length of both blocks.

The integration of these signals, as shown in Appendix A, and their relationship with the number and type of protons that each monomeric unit had allowed for an estimation of the composition of each triblock copolymer, as found in the literature [41]. Likewise, through calculations of the molecular weight (M*_v_*) of the PAM block, by viscometry, and relating it to the estimated composition, it was possible to estimate the molecular weight of both blocks. The results, as shown in Table 2, show that the feeding ratios were very close to the estimated final copolymers’ composition. This guarantees that the RAFT polymerization process occurred efficiently. Moreover, the PAM-*b*-PNIPAM-*b*-PAM3 triblock copolymer exhibited the highest viscous molecular weight compared with the other copolymers.

Due to the different chemical compositions of these triblock copolymers, their self-assembling behavior in solution is different from each other.

Amphiphilic block copolymers, either diblocks or triblocks, in aqueous solution tend to self-assemble into interesting morphologies that minimize the interaction of the hydrophobic block and water molecules and expose the water-affine segments (i.e., hydrophilic block). These varied morphologies are strongly related to the volume fraction of each block [42,43]. In fact, phase diagrams, which show the impact of the size of the hydrophobic block on these morphologies, are commonly used to design amphiphilic macromolecules with particular structural characteristics to obtain arrangements with specific sizes and morphologies depending on their potential application [44].

The triblock copolymers prepared in this work showed a similar behavior to the amphiphilic copolymers. At a temperature below the LCST, these copolymers showed a good affinity with water. However, when this temperature is exceeded, as in the Turkevich reaction, the PNIPAM block becomes hydrophobic, an entropically effect driven by the increase in temperature [45], while the PAM block maintains its hydrophilic character. Thus, under these conditions these triblock copolymers are considered amphiphilic macromolecules. Hence, the size of the blocks is a fundamental factor for the type of morphology adopted in the self-assembly process. Therefore, each triblock copolymer has a different impact on the formation of nanoparticles, since the environment during nanoparticle nucleation and growth is different with each block copolymer. These changes in the synthesis are evidenced by the different colors that these nanocomposites exhibit in solution, which is indicative of different sizes, morphologies, or nanoparticle concentrations. Analyzing these solutions by UV-Vis, Figure 4, it was shown that the composites displayed a plasmon band with an analogous shape and maximum value (520 nm). This implies that the gold nanoparticles were homogeneous in size and shape. Therefore, it was suggested that the main difference between these nanocomposites is related to the nanoparticle concentration. The triblock copolymer with a higher proportion of PAM, PAM-b-PNIPAM-b-PAM1, does not favor self-assembling structures with significant sizes, since it has a very small hydrophobic block. Thus, this macromolecule might tend to adopt a coil-like structural characteristic of the PAM homopolymer, despite the presence of the PNIPAM block and the thiocarbonate group. This conformation favors the copolymer to interact with sodium citrate molecules by hydrogen bonds, limiting the interaction of sodium citrate with the metallic precursor and even competing with these molecules in the gold reduction process, since the PAM block acts as a good reducing agent for gold [34]. Hence, the process of the reduction and formation of nanoparticles is highly affected. This impact on the process is observed since the solution, which has a yellow color due to the presence of the transfer agent, does not vary its color significantly, which is to be expected when there is a high concentration of gold nanoparticles, since they exhibit a characteristic ruby red color. In contrast, the copolymers PAM-*b*-PNIPAM-*b*-PAM2 and PAM-*b*-PNIPAM-*b*-PAM3, despite having a higher viscous molecular weight, do not affect the process of the reduction and formation of gold nanoparticles, only the stability and the size of the self-assemblies are affected [46]. This is because, as we mentioned above, the PAM block is the one that affects the reduction process that promotes the formation of nanoparticles. Thus, as the hydrophobic segment of PNIPAM increases, the macromolecules tend to shrink and assemble into structures that limit the interaction of the PAM block with metal precursors and sodium citrate molecules, so the Turckevich reaction occurs efficiently. Therefore, these nanocomposites have a high nanoparticle concentration, displaying a deeper reddish color. 

Likewise, the triblock copolymers have a strong impact on the nanoparticles’ dispersion into the polymeric chains. According to Figure 5, the distributions obtained by DLS indicate that the size of the nanocomposites’ assembly in solution increased when the copolymer had a higher PAM block composition. This is due to the fact that PAM has a strong affinity for gold nanoparticles thanks to the primary amide present in its structure. Therefore, the AuNPs are attached to the polymer chains, forming an aggregate with the polymer. In contrast, the copolymers with a considerable composition of the PNIPAM block have a low affinity for these nanostructures due to the considerable steric hindrance of its secondary amide. Then, the nanostructures could mostly be found isolated in the solution and not forming aggregates in combination with the copolymers.

In addition to the impact that the triblock copolymers had on the nanoparticles’ formation and dispersion in solution, the thermo-optical response of the nanocomposites was also strongly influenced by them. Likewise, the fact that the solutions were colorful, thanks to the presence of the AuNPs, makes it easy to follow any optical or conformation changes. The optical analyses by turbidimetry were made at room temperature and at 36 °C (T > LCST), as shown in Table 3. Commonly, the LCST of the PNIPAM is close to 33 °C. However, it has been shown that the combination of this with hydrophilic blocks, such as PAM, can increase the temperature of the LCST [47]. T is why 36 °C was selected to carry out this analysis. In addition, this type of nanocomposite is also considered for use as biosensors, and so this temperature is ideal, because it is close to the body’s temperature.

The opacity of PAM-*b*-PNIPAM-*b*-PAM-Au1 and PAM-*b*-PNIPAM-*b*-PAM-Au2 decrease with an increase in temperature to 36 °C. This suggests that due to the existence of a PAM block with a considerable size, the formation of thermally reversible hydrogen bonding at these conditions is favored. In other words, the hydrophilic character of these materials improves with increasing temperature, so their transition temperature is no longer considered as the LSCT but as the upper critical solution temperature (USCT) [48,49].

In contrast, PAM-*b*-PNIPAM-*b*-PAM-Au3 exhibits an LCST, since its opacity increases with an increasing temperature. This implies that its hydrophobicity increases at 36 °C by the self-assembly process. At this temperature, above the LCST, the structure is ordered in such a way that the segments (isopropyl groups) with less water affinity are exposed [45]. Thus, the refractive index of the material is drastically modified, “clouding” the solution [50].

The self-assembly process above the LCST (36 °C) was studied by STEM. The nanocomposite PAM-b-PNIPAM-b-PAM-Au1, as shown in Figure 6a, did not show any structural arrangement of the polymer at this temperature. In fact, this sample showed structural continuity. In the nanocomposite PAM-b-PNIPAM-b-PAM-Au2, the structural continuity was mostly maintained, as shown in Figure 5b. However, a few areas of the sample showed slight structural irregularities (inset in Figure 6b), which suggests that the size of the PNIPAM block does not favor a significant self-assembly. 

PAM-b-PNIPAM-b-PAM-Au3, in contrast with the other nanocomposites, showed structural arrangements due to the self-assembly process throughout the sample, as shown in Figure 6c. This is because the triblock copolymer is composed mostly of the PNIPAM block.

## 5. Conclusions

Triblock copolymers with an ABA block sequence design were prepared in two steps using a symmetrical transfer agent during the RAFT polymerization. The variation of the length of each block was achieved by varying the feed of the macro-RAFT agent (PAM block) and the NIPAM monomer in the second stage of the process.

All triblock copolymers were used as assistants in the formation of gold nanoparticles, by the Turkevich method, generating nanocomposites. Likewise, when these nanocomposites, in solution, are subjected to an increase in temperature (RT to 36 °C), their optical response varies depending on the triblock copolymers’ chemical composition. In fact, two of these materials exhibit a UCST, while the composite with the highest composition in the PNIPAM block maintains an LCST. This structural and optical versatility enhances the use of these materials in the development of new temperature monitoring devices.

## Figures and Tables

**Figure 1 polymers-15-01963-f001:**
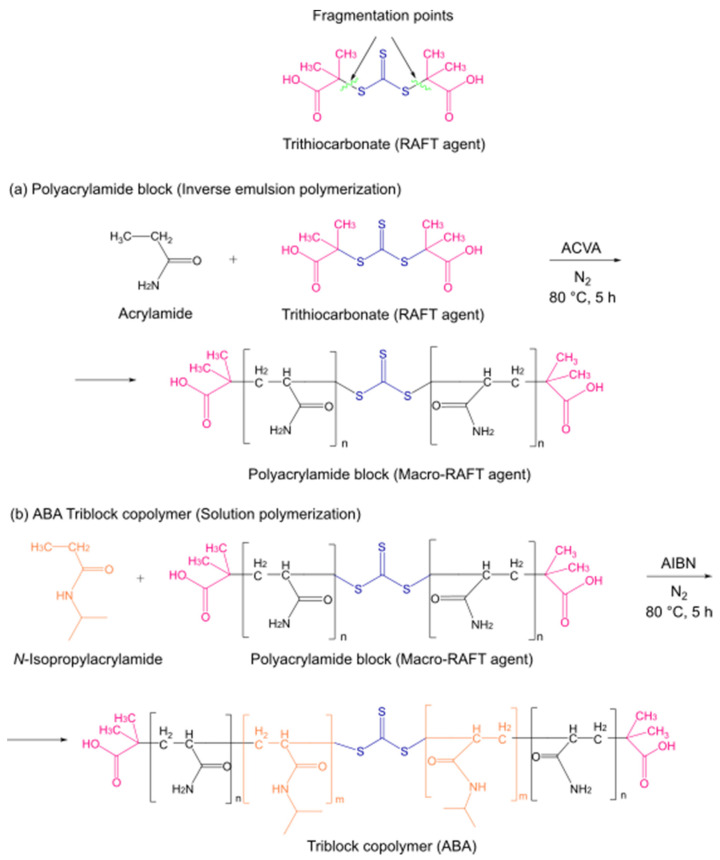
Representation of the two-step synthesis of triblock copolymers via thermally induced reversible addition–fragmentation chain-transfer polymerization (RAFT): (**a**) synthesis of the PAM block or macro-RAFT agent; (**b**) synthesis of the ABA triblock copolymers (PAM-*b*-PNIPAM-*b*-PAM).

**Figure 2 polymers-15-01963-f002:**
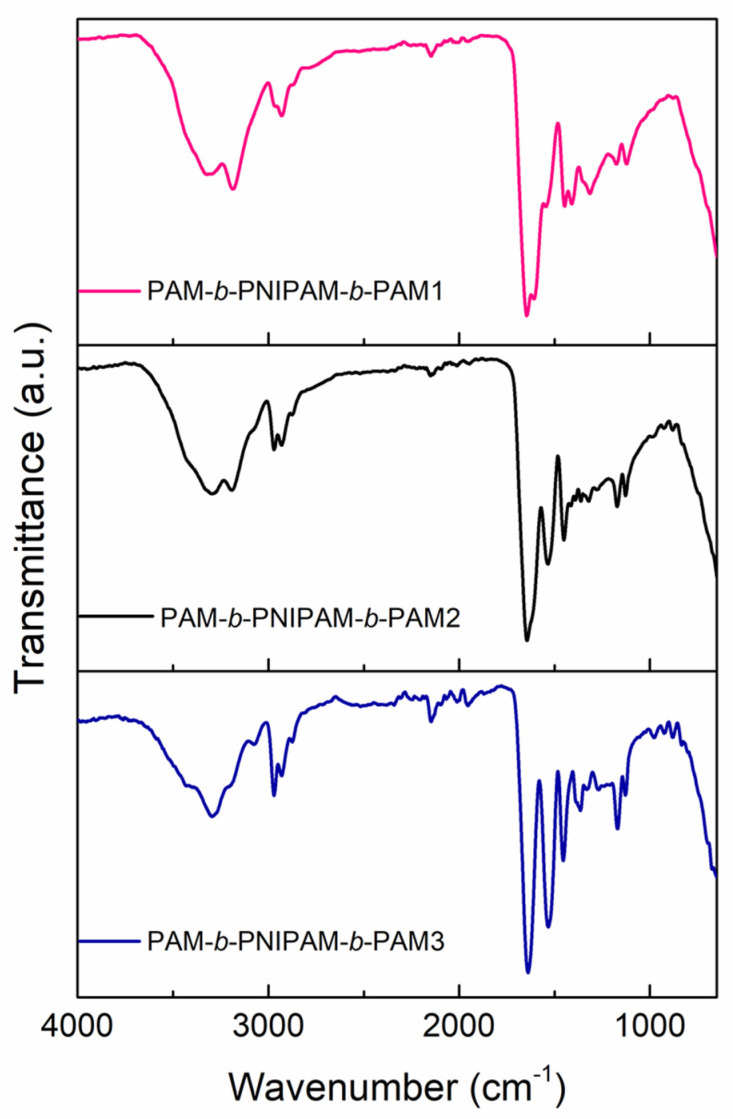
FT-IR spectra of the triblock copolymers: PAM−*b*−PNIPAM−*b*−PAM1, PAM−*b*−PNIPAM−*b*−PAM2, and PAM−*b*−PNIPAM−*b*−PAM3.

**Figure 3 polymers-15-01963-f003:**
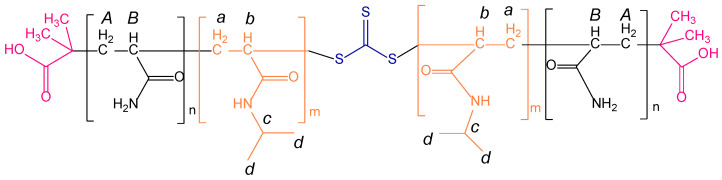
^1^H NMR spectra of triblock copolymers prepared by varying the macro-RAFT agent and NIPAM monomer feed. The chemical shifts of the PAM and PNIPAM blocks are represented by uppercase and lowercase letters, respectively. The PAM-*b*-PNIPAM-*b*-PAM2 and PAM-*b*-PNIPAM-*b*-PAM3 spectra show a magnification at 2.5–1.2 ppm to observe the variation of the intensity of the signals.

**Figure 4 polymers-15-01963-f004:**
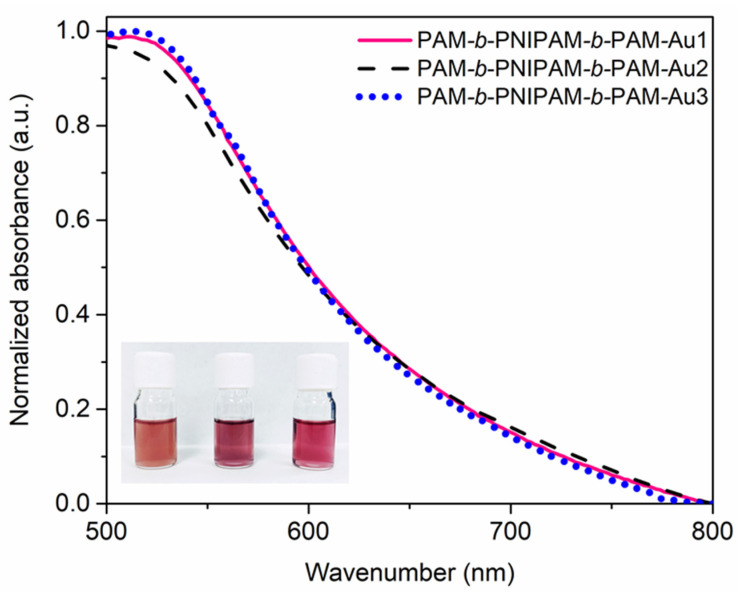
UV-Vis spectra of the nanocomposites in solution. The solutions display different colors associated with the impact of each copolymer on the nanoparticles’ formation. Inset image, from left to right: PAM-*b*-PNIPAM-*b*-PAM-Au1, PAM-*b*-PNIPAM-*b*-PAM-Au2, and PAM-*b*-PNIPAM-*b*-PAM-Au3 solutions.

**Figure 5 polymers-15-01963-f005:**
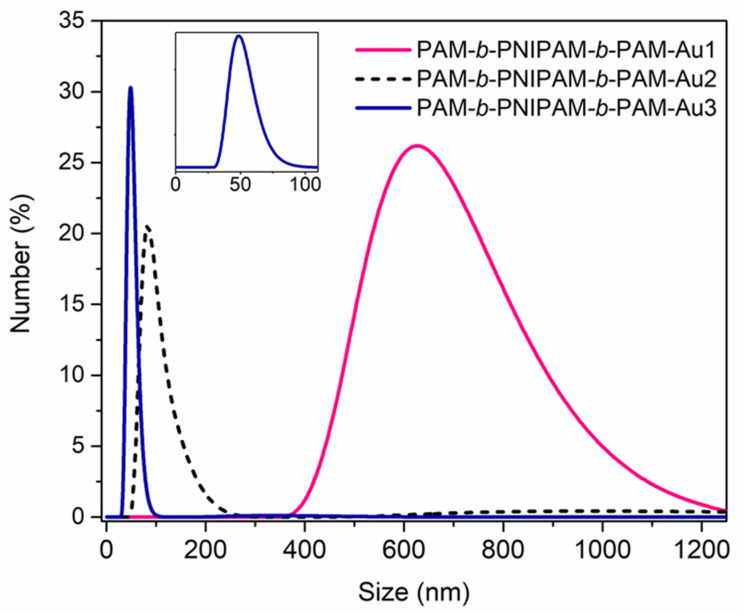
DLS number–size distribution of the nanocomposites in solution. By increasing the length of the size of the PAM block and decreasing the PNIPAM’s block size, a larger size distribution is observed. This might indicate the formation of aggregates between both components, the copolymer and the gold nanoparticles, due to the good affinity between PAM and AuNPs.

**Figure 6 polymers-15-01963-f006:**
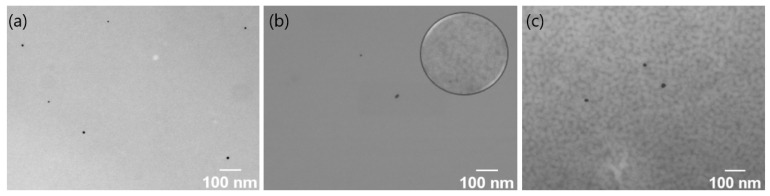
STEM micrographs of the nanocomposites prepared by drying at 36 °C: (**a**) PAM-*b*-PNIPAM-*b*-PAM-Au1; (**b**) PAM-*b*-PNIPAM-*b*-PAM-Au2; (**c**) PAM-*b*-PNIPAM-*b*-PAM-Au3. It is observed that by increasing the length of the PNIPAM block, the self-assembly process is favored, generating structural arrangements.

**Table 1 polymers-15-01963-t001:** Labels of the triblock copolymers and nanocomposites prepared.

Macro-RAFT Agent and Monomer Feed (%)		Triblock Copolymer (Labels)	Nanocomposite Labels
PAM	NIPAM		
80	20	PAM-*b*-PNIPAM-*b*-PAM1	PAM-*b*-PNIPAM-*b*-PAM-Au1
50	50	PAM-*b*-PNIPAM-*b*-PAM2	PAM-*b*-PNIPAM-*b*-PAM-Au2
20	80	PAM-*b*-PNIPAM-*b*-PAM3	PAM-*b*-PNIPAM-*b*-PAM-Au3

**Table 2 polymers-15-01963-t002:** Comparison between the macro-RAFT agent (PAM) and the NIPAM monomer feed, the estimated copolymers composition, and the calculated viscous molecular weight (Mv) of each block.

	Macro-RAFT Agent and Monomer Feed (%)		Estimated Triblock Copolymers’ Composition (%)		Viscosity Average Molecular Weight (g/mol) (M_v_)	
	PAM	NIPAM	PAM	PNIPAM	PAM	PNIPAM
PAM-*b*-PNIPAM-*b*-PAM1	80	20	84	16	3594	1018
PAM-*b*-PNIPAM-*b*-PAM2	50	50	42	58	3594	7808
PAM-*b*-PNIPAM-*b*-PAM3	20	80	15	85	3594	32,025

**Table 3 polymers-15-01963-t003:** Turbidimetry values obtained at room temperature (25 °C) and at a temperature above the LCST (36 °C).

	Turbidity (NTU)	
	25 °C	36 °C
PAM-*b*-PNIPAM-*b*-PAM-Au1	240	165
PAM-*b*-PNIPAM-*b*-PAM-Au2	185	124
PAM-*b*-PNIPAM-*b*-PAM-Au3	398	1144

## Data Availability

Not applicable.

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
