# Peer review of "Reversible Thermo-Optical Response Nanocomposites Based on RAFT Symmetric Triblock Copolymers (ABA) of Acrylamide and N-Isopropylacrylamide and Gold Nanoparticles"

_polymers, 2023, doi:10.3390/polym15081963_

Round 1

Reviewer 1 Report (Previous Reviewer 2)

Dear Author,

Thanks for incorporating changes as suggested. 

Author Response

Thank you for your comments on the manuscript.

Reviewer 2 Report (Previous Reviewer 1)

The authors discussed the synthesis of triblock copolymers with PNIPAM components and its application for thermo-optical response. NMR, DLS, STEM and etc were used for the characterizations. However, there are some issues/problems needed to be addressed:

  1. What is the novelty point of this study? Neither the synthesis of triblock copolymers or the application of thermo-optical response for PNIPAM is new to the society. It has been widely studied by peers and I do not see the new points claimed by the authors.
  2. Following by previous point, the main point claimed by the authors is the different behaviors of the copolymers in solution due to variations in compositions. However, the authors did not discuss it deeply and just reveal the observed phenomenon. I would suggest the authors try to study it more deeply.
  3. The authors claimed the different behaviors were due to the composition variations. However, given the very different Mv of the polymers, I would expect this is another factor that the authors need to consider and discuss.
  4. Only the viscosity average molecular weights were provided within this study. I wonder why the authors not use SEC (GPC) to characterize the synthesized copolymers?
  5. For the NMR Figures, I noticed there are some negative peaks within the results, which is unreasonable to me. Please make sure the data processing is correct. In addition, please assign the peaks accordingly otherwise there is very limited information can be obtained from the results.
  6. I can see the amount of work has been done by the authors. And I appreciate the efforts. However, I would suggest the authors make a plan to study this topic deeply and find the novelties within this study. Given current manuscript, I have to give a rejection recommendations.

Author Response

1. What is the novelty point of this study? Neither the synthesis of triblock copolymers or the application of thermo-optical response for PNIPAM is new to the society. It has been widely studied by peers and I do not see the new points claimed by the authors.

Answer:

Despite the wide range of block copolymers synthesized and reported elsewhere, few reports have described the synthesis of PAM-b-PNIPAM copolymers for thermosensitive materials through reversible-deactivation radical polymerization (RDRP) techniques. For example, Perrier et al. reported an aqueous RAFT polymerization for the preparation of multiblock copolymers with acrylamides and their derivatives [1]. Broekhius et al. reported PAM-b-PNIPAM block copolymers via ATRP that exhibited interesting rheological properties, indicating a potential application of the purchased polymers in improved oil recovery processes [2-4]. Recently, Jimenez-Regalado et al. [5] reported the effect of N-isopropylacrylamide thermoresponsive blocks on the rheological properties of water-soluble thermoassociative copolymers synthesized via RAFT polymerization. However, unlike this report, in our work the first block was synthesized by using inverse emulsion polymerization.

Thus, the novelty of our work lies in obtaining controlled block copolymers of acrylamide and isopropylacrylamide, as well as their application as substrates in the synthesis of nanoparticles or nanocomposites and their thermo-optical response. Therefore, our manuscript is according with the scope of the special issue.

[1] Martin L, Gody G, Perrier S (2015) Preparation of complex multiblock copolymers via aqueous RAFT polymerization at room temperature. Polym Chem 6:4875–4886

[2] Wever DAZ, Raffa P, Picchioni F, Broekhuis AA (2012) Acrylamide homopolymers and acrylamide-N-isopropylacrylamide block copolymers by atomic transfer radical polymerization in water. Macromolecules 45:4040–4045

[3] Wever DAZ, Polgar LM, Stuart MCA, Picchioni F, Broekhuis AA (2013) Polymer molecular architecture as a tool for controlling the rheological properties of aqueous polyacrylamide solutions for enhanced oil recovery. Ind Eng Chem Res 52:16993–17005

[4] Raffa P, Broekhuis AA, Picchioni F (2016) Polymeric surfactants for enhanced oil recovery: a review. J Pet Sci Eng 145:723–733

[5] Díaz-Silvestre SD, St Thomas C, Maldonado-Textle H, Rivera-Vallejo C, Diaz de León-Gómez R, Jiménez-Regalado EJEffect of N-isopropylacrylamide thermoresponsive blocks on the rheological properties of water-soluble thermoassociative copolymers synthesized via RAFT polymerization, Colloid Polym Sci (2018) 296:1699–1710.

2. Following by previous point, the main point claimed by the authors is the different behaviors of the copolymers in solution due to variations in compositions. However, the authors did not discuss it deeply and just reveal the observed phenomenon. I would suggest the authors try to study it more deeply.

Answer:

We agree with the reviewer for this reason we modified the discussion on how these triblock copolymers affect the formation process of nanoparticles. We added into the manuscript:

“Amphiphilic block copolymers, either diblocks or triblocks, in aqueous solution tend to self-assemble into interesting morphologies that minimize interaction of the hydrophobic block to water molecules and expose the water-affine segments (i.e., hydrophilic block). These varied morphologies are strongly related to the volume fraction of each block [42, 43]. In fact, phase diagrams which show the impact of the size of the hydrophobic block on these morphologies are commonly used to design amphiphilic macromolecules with particular structural characteristics to obtain arrangements with specific sizes and morphologies depending on their potential application [44].

The triblock copolymers prepared in this work show a similar behavior to the amphiphilic copolymers. At a temperature below the LCST, these copolymers show a good affinity with water. However, when this temperature is exceeded, as in the Turkevich reaction, the PNIPAM block becomes hydrophobic, an entropically effect driven by the increase in temperature [45], while the PAM block maintains its hydrophilic character. Thus, at these conditions these triblock copolymers are considered as amphiphilic macromolecules. Hence, the size of the blocks is a fundamental factor for the type of morphology adopted in the self-assembly process. Therefore, each triblock copolymer has a different impact in the nanoparticles formation since the environment during nanoparticle nucleation and growth is different with each block copolymer.”

3. The authors claimed the different behaviors were due to the composition variations. However, given the very different Mv of the polymers, I would expect this is another factor that the authors need to consider and discuss.

Answer:

We thank the reviewer for bringing this up. As shown in the rewritten discussion, it has been proven that the molecular weight of the copolymer impacts on the stability and size of the array obtained by self-assembly. However, in this approach the PAM block is the one that affects the reduction process that promotes the formation of nanoparticles, therefore this parameter is out of scope for our work.

4. Only the viscosity average molecular weights were provided within this study. I wonder why the authors not use SEC (GPC) to characterize the synthesized copolymers?

Answer:

Many thanks for the comment, we appreciate the suggestion. We know that one of the most important characterizations for polymeric materials is the determination of their molecular weight distribution and their polydispersity index (PDI). However, due to the publication time for this special issue and the limited access to these equipment, it was difficult to perform this GPC characterization. Nevertheless, we consider that the analysis of viscous molecular weight and its relationship with the compositions estimated by 1H NMR  give an accurate value on the different lengths of the blocks since they are methodologies commonly used in various works (https://doi.org/10.1016/j.polymer.2022.125207, DOI: 10.1515/psr-2019-0086, https://doi.org/10.1295/polymj.19.297, https://doi.org/10.1002/app.50850).

5. For the NMR Figures, I noticed there are some negative peaks within the results, which is unreasonable to me. Please make sure the data processing is correct. In addition, please assign the peaks accordingly otherwise there is very limited information can be obtained from the results.

Answer:

The 13C NMR spectra provided complementary information on obtaining the copolymers, however they have been removed from the supplementary material, since the 1H NMR and FT-IR spectra confirm the obtaining of the macromolecular structures. However, in response to the reviewer's request, the assignment of the signals in the NMR spectrum for the case of PAM-b-PNIPAM-b-PAM2: 180 ppm and 175 ppm, corresponding to the C=O of polyacrylamide and poly(N-isopropylacrylamide), respectively;  42 ppm and 35 ppm, related to the carbon CH and CH2 moieties of the resulting backbone of the polyacrylamide and poly(N-isopropylacrylamide), finally, the signal at 20 ppm related at the methyl groups (CH3) corresponding to NIPAM.

6. I can see the amount of work has been done by the authors. And I appreciate the efforts. However, I would suggest the authors make a plan to study this topic deeply and find the novelties within this study. Given current manuscript, I have to give a rejection recommendations.

Answer:

We appreciate the reviewer's comments and suggestions. In the current version of the manuscript, we have included new paragraphs that complement the novelty and discussion of the extensive experimental work.

Reviewer 3 Report (New Reviewer)

The manuscript, entitled "Reversible thermo-optical response nanoparticles and RAFT triblock copolymer (ABA) of acrylamide and N-Isopropylacrylamide", presents an interesting work on using thermo-responsive ABA copolymers for stabilizing nanoparticles and modulating the temperature response of these nanoparticles. There are a few questions that the authors could address before it gets accepted:

1. What would happen if you use homopolymers (acrylamide or N-isopropylacrylamide) to do the same type of experiment? How about random copolymers using these monomers? These experiments would be helpful for understanding the contributions of different polymer species and architectures.

2. Do you have a temperature dependent turbidity curve? Does triblock copolymer composition affect the transition temperature?

Author Response

1. What would happen if you use homopolymers (acrylamide or N-isopropylacrylamide) to do the same type of experiment? How about random copolymers using these monomers? These experiments would be helpful for understanding the contributions of different polymer species and architectures.

Answer:

Many thanks, we appreciate the suggestion. Previously, our work group showed how polyacrylamide and hydrolyzed polyacrylamide affect the nucleation and growth process of gold nanoparticles, which is why this monomer was selected to build the triblock copolymers shown in this work (https://doi.org/10.3390/ma15238557). The action of the PNIPAM, on the other hand, has already been widely studied (https://doi.org/10.1021/acsnano.5b04083, https://doi.org/10.1021/ja038544z, https://doi.org/10.1021/la0363938, https://doi.org/10.1002/cphc.201800891, https://doi.org/10.1016/j.colsurfa.2022.128409). That is why the novelty of this work was to build a symmetrical triblock copolymer with an ABA sequence from both monomers.

2. Do you have a temperature dependent turbidity curve? Does triblock copolymer composition affect the transition temperature?

Answer:

We thank the reviewer for bringing this up. We have included the next paragraph, and a reference [47],  to show that indeed the LCST value varies by the composition of the copolymer and the main reason why we selected this temperature and did not perform a turbidimetry curve.

“Commonly, the LCST of the PNIPAM is close to 33 °C. However, it has been shown that the combination of this with hydrophilic blocks, such as PAM, can increase the temperature of LCST. That is why 36 °C was selected to carry out this analysis. In addition, it is also contemplated to be able to use this type of nanocomposites as biosensors, so that temperature is ideal because it is close to body temperature.”

Round 2

Reviewer 2 Report (Previous Reviewer 1)

Thank you for providing the updated manuscript and responding to my concerns. It is clear that the authors have put a lot of effort into revising the manuscript. However, I still have some concerns and questions regarding the updated version:

1. Although the authors provided relevant studies to address my concerns about novelty, I still have some reservations. Some of the cited studies involve unique structures, detailed kinetic or application studies, which contribute to their novelty. I would be expecting to see some more comprehensive studies and discussions by the authors.

2. The authors stated that inverse emulsion polymerization plays an important role in the study, but this should be discussed further.

3. The thermo-optical behaviors of the systems were emphasized by the authors, and I would be expecting to see some more discussion regarding the thermo-optical behaviors. For instance, instead of the turbidimetry values at 2 specific temperatures, curves would be more informative. Meanwhile, I think the transition from LSCT to USCT worths some further discussions.

4.There are some minor formatting issues, such as using "Figure" instead of "figure" in the content."

Author Response

1. Although the authors provided relevant studies to address my concerns about novelty, I still have some reservations. Some of the cited studies involve unique structures, detailed kinetic or application studies, which contribute to their novelty. I would be expecting to see some more comprehensive studies and discussions by the authors.

Answer

We appreciate the reviewer's comment. As authors committed to improving this work, each of the comments that you and the other reviewers have kindly provided us to enrich our manuscript have been taken into consideration and carried out. Prompted by this, we consider that the characterization techniques made and the discussion provided throughout the main text are adequate for the scope of this work

2. The authors stated that inverse emulsion polymerization plays an important role in the study, but this should be discussed further.

Answer

We appreciate the comment and agree with the reviewer that we should have included information about emulsion polymerization.  Therefore, we have included the next paragraph: “For the first part, i.e., the formation of the first block, the inverse emulsion polymerization method was used [34]. This technique takes place in a heterogeneous milieu constituted by two immiscible phases; namely, an organic solvent that constitutes the continuous phase and the aqueous dispersed phase. The continuous phase contains the initiator as well as the transfer agent, while the aqueous phase dissolves the monomer. During synthesis the interaction between both phases is facilitated by a surfactant, creating micelles with the dispersed phase in the interior. Thus throughout the reaction, all the different stages of this polymerization process, including chain growth, occur within those micelles.“

3. The thermo-optical behaviors of the systems were emphasized by the authors, and I would be expecting to see some more discussion regarding the thermo-optical behaviors. For instance, instead of the turbidimetry values at 2 specific temperatures, curves would be more informative. Meanwhile, I think the transition from LSCT to USCT worths some further discussions.

Answer

We appreciate the comment. Within most of the manuscript main text, we have discussed extensively the optical response of these materials, especially the one with the great proportion of the thermosensitive block (PNIPAM). Besides, we have included the next paragraph, and a reference [47], to show that indeed the LCST value varies by the composition of the copolymer and the main reason why we selected this temperature and did not perform a turbidimetry curve.

“Commonly, the LCST of the PNIPAM is close to 33 °C. However, it has been shown that the combination of this with hydrophilic blocks, such as PAM, can increase the temperature of LCST [47]. That is why 36 °C was selected to carry out this analysis. In addition, it is also contemplated to be able to use this type of nanocomposites as biosensors, so that temperature is ideal because it is close to body temperature.”

4. There are some minor formatting issues, such as using "Figure" instead of "figure" in the content."

Answer

We appreciate the comment, these minor formatting issues were addressed.

Reviewer 3 Report (New Reviewer)

The authors have addressed my points from last review properly. Although the authors did not add additional data as I suggested, the evidence from literature is adequate for answering my questions. I think it is a good one to be accepted by MDPI Polymers.

Author Response

we appreciate the reviewer's comments

Round 3

Reviewer 2 Report (Previous Reviewer 1)

Thank you for the updated manuscript. I do not have further questions regarding it. 

This manuscript is a resubmission of an earlier submission. The following is a list of the peer review reports and author responses from that submission.

Round 1

Reviewer 1 Report

The authors discussed the synthesis and application of triblock copolymers for the application of thermo-optical response nanocomposites. Characterizations include NMR, UV-Vis, DLS and STEM were provided to support the research. However, there are still some issues/questions needed to be addressed.

  1. One very important thing was not fully explained within the introduction as: why do the authors choose triblock polymers? What is the difference if diblock polymer instead of triblock polymer is used within this study?
  2. For the synthesis part, mol number should also be provided.
  3. For the triblock synthesis, I can see the potential coupling of the RAFT agent during the polymerization. Have the authors characterize that? Please provide some more details regarding this potential issue.
  4. GPC (or SEC) results should be provided to understand the Mw distribution as well as the conversion of the monomers. And the molecular information as well as the PDI should be provided.
  5. For Figure 2 NMR results, the integrated H number for each peak should be provided. Otherwise there are very little information can be provided within the Figure. Table 1 is helpful but some more detailed information should be provided within the Figure directly. Also, please provide C-13 NMR results within the supporting information.
  6. For the results provided in Table 1, I would assume it would be highly impacted by the residual unreacted monomers as well as the potential polymerized polymer without the RAFT agent. Please comment on it.
  7. It is really hard to read Figure 3. Please use more identifiable colors to label the them. Also, to me, the most different area is between 500 - 600nm. Probably the authors can zoom in the area to make it better for authors to understand.
  8. Please include the reference as: https://doi.org/10.1021/acs.macromol.8b02366; https://doi.org/10.1039/D0PY01498B; https://doi.org/10.1021/acs.langmuir.6b00284

Reviewer 2 Report

Dear Author,

The article describes composite materials of gold and triblock polymers containing polynipam and polyacrylamide which was synthesized by RAFT polymerization technique. Here are my comments:

(1) My major concern is the novelty of this work. This type of work has been done before where scientists show the change in optical properties with a change in the thickness of the polymer surrounding the metal nanoparticle. How your study is different than others?

(2) The author should describe the method used in experiments, especially Turckevich method.

(3) No control was run or missing

(4) In RAFT process how NIPAM comes in between acrylamide. Please add a paragraph the process in results and discussion for the readers.

Reviewer 3 Report

Comments

Summary

This article describes a RAFT-mediated synthesis of a thermoresponsive triblock copolymer with different copolymer compositions followed by the synthesis of gold nanoparticles by following Turkevich reaction. Stabilization of Au-NPs using triblock copolymer has been monitored too. There are plenty of literatures available where Au-NPs were stabilized with Poly(NIPAAm)-based copolymer and their thermoresponsive behavior was observed. I could not find any novelty in this work and based on this I am not recommending this manuscript to be published in “Polymers”. My comments are summarized below-

Major Comments

1.      Authors should include relevant references in the introduction section to highlight the novelty of their work compared to others. There are literatures available where Au NPs have been synthesized and stabilized subsequently with PNIPAAm-based copolymers. Few of them are mentioned below-

·        Maji, S., Cesur, B., Zhang, Z., De Geest, B. G., & Hoogenboom, R. (2016). Poly (N-isopropylacrylamide) coated gold nanoparticles as colourimetric temperature and salt sensors. Polymer Chemistry, 7(9), 1705-1710.

·        Liu, Y., Tu, W., & Cao, D. (2010). Synthesis of Gold Nanoparticles Coated with Polystyrene-block-poly (N-isopropylacrylamide) and Their Thermoresponsive Ultraviolet− Visible Absorbance. Industrial & Engineering Chemistry Research, 49(6), 2707-2715.

2.      Author should rewrite both the “Abstract” and “Introduction” section by mentioning the novelty of their work.  

3.       Synthesized polymers should be characterized properly using GPC and FTIR analyses along with the proton NMR.

4.      In proton NMR, integration values should be included in the resonance spectra so that readers could understand the composition of the triblock copolymer out of the NMR.

5.      Characterization section should contain the detail of all techniques such as concentration of the polymer solution used, polymers casting procedure, TEM grid characteristics etc.

6.      STEM picture doesn’t designate the shape and diameter of the organic-inorganic hybrid composite. I am assuming that composite has a core-shell type of structure. From the picture it seems to be a continuous film.

7.      Reaction scheme should include all reaction conditions such as temperature, reaction environment etc.

8.      From Figure 5, it seems that PAMb80-PNIPAMb20-Au is showing a very broad distribution which is not the case for the rest of two. The reason of this should be explained clearly.

Minor Comments

1.      English should be refined throughout the manuscript.